# New Guaiane-Type Sesquiterpenoids Biscogniauxiaols A–G with Anti-Fungal and Anti-Inflammatory Activities from the Endophytic Fungus *Biscogniauxia Petrensis*

**DOI:** 10.3390/jof9040393

**Published:** 2023-03-23

**Authors:** Long Han, Wen Zheng, Sheng-Yan Qian, Ming-Fei Yang, Yong-Zhong Lu, Zhang-Jiang He, Ji-Chuan Kang

**Affiliations:** 1College of Life Sciences, Guizhou University, Guiyang 550025, China; 2Engineering and Research Center for Southwest Bio-Pharmaceutical Resources of National Education Ministry of China, Guizhou University, Guiyang 550025, China; 3Guizhou Institute of Technology, School of Food and Pharmaceutical Engineering, Guiyang 550003, China

**Keywords:** endophytic fungus, *Biscogniauxia petrensis*, sesquiterpenoids, anti-fungal activity, anti-inflammatory activity, multidrug resistance reversal activity

## Abstract

Seven undescribed guaiane-type sesquiterpenoids named biscogniauxiaols A–G (**1**–**7**) were isolated from the endophytic fungus *Biscogniauxia petrensis* on *Dendrobium orchids.* Their structures were determined by extensive spectroscopic analyses, electronic circular dichroism (EC) and specific rotation (SR) calculations. Compound **1** represented a new family of guaiane-type sesquiterpenoids featuring an unprecedented [5/6/6/7] tetracyclic system. A plausible biosynthetic pathway for compounds **1**–**7** was proposed. The anti-fungal, anti-inflammatory and multidrug resistance reversal activities of the isolates were evaluated. Compounds **1**, **2** and **7** exhibited potent inhibitory activities against *Candida albicans* with MIC values ranging from 1.60 to 6.30 μM, and suppressed nitric oxide (NO) production with IC_50_ ranging from 4.60 to 20.00 μM. Additionally, all compounds (100 μg/mL) enhanced the cytotoxicity of cisplatin in cisplatin-resistant non-small cell lung cancer cells (A549/DDP). This study opened up a new source for obtaining bioactive guaiane-type sesquiterpenoids and compounds **1**, **2**, and **7** were promising for further optimization as multifunctional inhibitors for anti-fungal (*C*. *albicans*) and anti-inflammatory purposes.

## 1. Introduction

Endophytic fungi are considered a source of new bioactive natural products with the potential to become new drugs [1]. Meanwhile, as the main sources of active sesquiterpenoids with untapped structural diversity and multiple biological activities, endophytic fungi have attracted increasing attention from organic chemists and pharmacologists [2]. Guaiane-type sesquiterpenoids are typical bicyclic sesquiterpenes possessing a [5/7] fused ring skeleton, and [5/7/5], [5/6/5], [5/7/6], [5/5/7] tricyclic chemical scaffolds, [5/5/5/6], [5/5/6/5] tetracyclic systems and [5/5/6/5/5] pentacyclic system were afforded via oxidation and cyclization reactions [3,4,5]. These compounds usually exhibit potent bioactivities, including anti-fungal, anti-inflammatory, anticancer and multidrug resistance (MDR) reversal activities [6,7,8,9]. However, plants are the main source of guaiane-type sesquiterpenoids, such as *Daphne genkwa*, *Pogostemon cablin*, *Xylopia vielana*, *Stellera chamaejasme*, *Artemisia zhongdianensis*, *Curcuma wenyujin* [10,11,12,13,14,15], and there are few reports in microorganisms, which restricts the development and utilization of such active substances.

In our endeavor to search for new bioactive compounds from the endophytic fungi of medicinal plants, the fungus *Biscogniauxia petrensis* MFLUCC 14-0151 isolated from the *Dendrobium orchids* was investigated [16], leading to the identification of one unusual iridoid biscogniauxiaol A (**1**), which possesses an unprecedented [5/6/6/7] tetracyclic system and six new [5/7] bicyclic guaiane-type sesquiterpenoids biscogniauxiaol B–G (**2**–**7**) (Figure 1). Herein, the details of the isolation, a discussion of their structural characterization, a plausible biosynthetic pathway for **1**–**7** and their bioactivities were reported.

## 2. Materials and Methods

### 2.1. General Experimental Procedures

The optical rotations and CD spectra were measured on Autopol VI in MeOH (Rudolph Research Analytical, Hackettstown, NJ, USA) and Chirascan CD spectrophotometer in MeOH (Applied Photophysics, Leatherhead, UK), respectively. The IR spectra via KBr pellets and UV spectra were recorded on Nicolet iS10 spectrometer (Thermo Fisher Scientific, Madison, WI, USA) and Shimadzu UV2401PC (Shimadzu, Kyoto, Japan), respectively. NMR spectra (500 MHz for ^1^H and 125 MHz for ^13^C) were recorded using Avance III 500 MHz equipment (Bruker, Bremerhaven, Germany). HRFABMS data and HRESIMS data were determined using Fast-atom-bombardment mass spectrometry DFS (Thermo Fisher Scientific, Madison, WI, USA) and Shimadzu LC/MS-IT-TOF mass instrument (Shimadzu, Kyoto, Japan), respectively. Column chromatography was conducted on silica gel (200–300 mesh, Qingdao Puke Abruption Materials Co., Ltd. Qingdao, China) and Sephadex LH-20 (Shanghai Yuanye Bio-Technology Co., Ltd. Shanghai, China). ODS column chromatography was performed using C18 silica gel (Fuji Silysia Chemical Ltd. Kasugai, Japan). TLC was carried out on precoated glass silica gel GF254 plates and compounds were visualized under UV light or via heating silica gel plates sprayed with 5% H_2_SO_4_/EtOH.

### 2.2. Fungal Material

The fungus *B. petrensis* MFLUCC14-0151 was isolated and identified by our research group [16], and preserved at the China General Microbiological Culture Collection Center (CGMCC 40341), Beijing.

### 2.3. Fermentation, Extraction and Isolation

Martin modified (MM) medium was inoculated with the aforementioned fungus and incubated in a constant-temperature incubator at 28 °C for 5 d to obtain seed culture. Fermentation was carried out in a conical flask (1 L) containing 200 g of rice and 150 mL of distilled water, and then autoclaving at 120 °C for 30 min. Each flask was inoculated with 10 mL of seed culture and incubated at 28 °C for two months.

The fermented cultures were extracted thrice with 2-fold volume of methanol and the organic solvent was evaporated to a small volume under vacuum, and then suspended in H_2_O (10 L) and partitioned successively with 20 L of ethyl acetate and n-Butanol, respectively. The ethyl acetate solution was evaporated under reduced pressure to afford a crude extract (56 g). The extract was separated into 6 fractions (Fr. A1–A6) via silica gel column chromatography using ethyl acetate/methanol (50:1, 25:1, 10:1, 5:1 and 1:1). Fraction A1 (9.825 g) was separated by ODS MPLC eluted with methanol/H_2_O (20:80, 40:60, 60:80, and 90:10) to afford 13 sub-fractions A1 (1–13). The fraction A1-1-8 (192 mg) was purified by silica gel column chromatography eluted with dichloromethane/ethyl acetate (1:1) to afford sub-fraction A1-1-8-1 (32 mg). Further purification of fraction A1-1-8-1 via ODS MPLC with H_2_O/acetone (90:10) yielded compound **1** (4.8 mg). Fraction A2 (10.73 g) was separated via ODS MPLC eluted with H_2_O/ethanol (20:80, *v*/*v*) to provide 10 sub-fractions A2 (1–10). Fraction A2-1 (189 mg) was purified by silica gel column chromatography eluted with petroleum ether/acetone (2:1) to yield compound **2** (20 mg) and compound **3** (10 mg). Separation of fraction A2-2 (819 mg) via a silica gel column with n-hexane/acetone/chloroform (2:2:1) and Sephadex LH-20 (methanol) yielded compound **4** (25 mg) and compound **5** (22 mg). Fraction A2-3 (1.2 g) was subjected to silica gel column with ethyl acetate/acetone/petroleum ether (1:1:1) and Sephadex LH-20 (methanol) to obtain compound **6** (10 mg) and compound **7** (8 mg).

Biscogniauxiaol A (**1**). Colorless solid;
$[\alpha {]}_{D}^{20}$ + 36.50 (*c* 0.20, MeOH); UV (MeOH) λ max (log ε) 203 (0.26), 275 (0.01); IR *ν* max 3438, 2967, 2929, 2872, 1722, 1620, 1488, 1372, 1275, 1224, 1170, 1097, 1075, 955, 855, 771 cm^−1^; ^1^H and ^13^C NMR data, see Table 1; HRFABMS ([M + H]^+^ Calcd for C_15_H_23_O_3_, 251.1642; found 251.1644).

Biscogniauxiaol B (**2**). Colorless solid;
$[\alpha {]}_{D}^{20}$ −2.33 (*c* 0.12, MeOH); UV (MeOH) λ max (log ε) 204 (0.46); IR *ν* max 3403, 2930, 2874, 1640, 1460, 1382, 1162, 1069, 1039, 907, 817 cm^−1^; ^1^H and ^13^C NMR data, see Table 1; (+)-HERMS ([M + Na]^+^ Calcd for C_15_H_28_O_3_Na, 279.1931; found 279.1927).

Biscogniauxiaol C (**3**). Colorless solid;
$[\alpha {]}_{D}^{20}$ −10.78 (*c* 0.10, MeOH); UV (MeOH) λ max (log ε) 202 (0.35), 218 (0.27); IR *ν* max 3414, 2930, 1641, 1457, 1383, 1264, 1083, 1049, 940, 887 cm^−1^; ^1^H and ^13^C NMR data, see Table 1; HERMS ([M + Na]^+^ Calcd for C_15_H_28_O_4_Na, 295.1880; found 295.1882).

Biscogniauxiaol D (**4**). Colorless solid;
$[\alpha {]}_{20}^{D}$ −1.88 (*c* 0.12, MeOH); UV (MeOH) λ max (log ε) 203 (0.34), 276 (2.02); IR *ν* max 3432, 2934, 2873, 1633, 1459, 1381, 1046, 939, 907 cm^−1^; ^1^H and ^13^C NMR data, see Table 2; HRESIMS ([M + Na]^+^ Calcd for C_15_H_28_O_4_Na, 295.1880; found 295.1870).

Biscogniauxiaol E (**5**). Colorless solid;
$[\alpha {]}_{D}^{20}$ −4.50 (*c* 0.12, MeOH); UV (MeOH) λ max (log ε) 203 (0.30), 223 (0.12), 275 (0.04); IR *ν* max 3403, 2956, 1634, 1464, 1380, 1335, 1273, 1192, 1169, 1142, 1087, 1049, 955, 925, 875, 823, 779 cm^−1^; ^1^H and ^13^C NMR data, see Table 2; HRESIMS ([M + Na]^+^ Calcd for C_15_H_28_O_4_Na, 295.1880; found 295.1878).

Biscogniauxiaol F (**6**). Colorless solid;
$[\alpha {]}_{D}^{20}$ −11.30 (*c* 0.10, MeOH); UV (MeOH) λ max (log ε) 203 (0.50), 221 (0.25), 280 (0.10), 348 (0.16); IR *ν* max 3436, 2925, 1636, 1456, 1383, 1318, 1164, 1112, 1060, 1034 cm^−1^; ^1^H and ^13^C NMR data, see Table 2; HRESIMS ([M + Na]^+^ Calcd for C_16_H_28_O_4_Na 307.1880; found 307.1873).

Biscogniauxiaol G (**7**). Colorless solid;
$[\alpha {]}_{D}^{20}$ −15.25 (*c* 0.20, MeOH); UV (MeOH) λ max (log ε) 202 (0.42), 271 (0.02); IR *ν* max 3417, 2924, 2869, 1638, 1455, 1380, 1066, 1031, 900 cm^−1^; ^1^H and ^13^C NMR data, see Table 2; HRESIMS ([M + Na]^+^ Calcd for C_15_H_28_O_3_Na, 277.1774; found 277.1764).

### 2.4. Quantum Chemical Calculation (ECD)

The systematic random conformational analyses were performed in the GMMX program by using a MMFF94 molecular force field, which afforded a few conformers each, with an energy cutoff of 10 kcal/mol to the global minima. All of the obtained conformers were further optimized using DFT at the B3LYP/6-31G(d) level in CH_3_OH by using Gaussian 09 software [17]. The optimized stable conformers were used for TDDFT [B3LYP/6-311G(2d,p)] computations, with the consideration of the first 20 excitations. The overall ECD curves were all weighted by the Boltzmann distribution. The calculated ECD spectra were subsequently compared with the experimental ones. The ECD spectra were produced by SpecDis 1.70.1 software [18].

### 2.5. Specific Rotation Calculation (SRC)

Pcmodel program (version 10.075) was used to generate conformers at the MMFF94 force field [18]. Then, the isomers were preoptimized with the molculs program (version 1.9.9) by invoking xTB at GFN2-xTB level [18,19,20,21]. The clusters were optimized at B3LYP/def2-TZVP level with ORCA program and ensured the optimized structures have no imaginary frequency. After all the Boltzmann population properties were obtained from the Gibbs free energy calculated at PWPB95/def2-QZVPP with thermal corrections in methanol with the SMD solvation model [22]. Next, the most populated conformations obtained were used for specific optical rotations calculation at the B3LYP/6-311 + g(2d,p) level with the Gaussian 09 program package and integrated according to Boltzmann weighting proportions [17].

### 2.6. Anti-Fungal Assay

The minimal inhibitory concentration (MIC) values of each compound against *Candida albicans* (336485) (BeNa Culture Collection, Henan, China), were determined using the broth microdilution method [23]. Briefly, the screened compounds were 2-fold serially diluted in cell suspensions in RPMI 1640 medium. Then, 100 μL aliquots were added to 96-well plates. The plates were incubated at 35 °C for 24 h. The commercial amphotericin B (AMB) and fluconazole (FCZ) were used as the positive controls. Zero visible growth was considered as the endpoint value according to the guidelines (M27-A3) (CLSI 2008) [24]. All experiments were carried out in triplicate.

### 2.7. Anti-Inflammatory Assay

RAW264.7 cells (BeNa Culture Collection, Henan, China) were cultured in Dulbecco’s Modified Eagle Medium (DMEM) supplemented with 10% Fasting Blood Sugar (FBS). To determine the anti-inflammatory activities of compounds **1**–**7**, the cells were pretreated with fresh DMEM medium (100 μL/well) containing the tested compounds at various final concentrations (0–100 μg/mL) for 2 h. Then, the lipopolysaccharide (LPS, 1 μg/mL) was added and cultured for another 24 h. NO production in the supernatant was assessed using Griess reagents [4]. The absorbance at 540 nm was measured on a microplate reader. Indomethacin was used as the positive control. Meanwhile, the viability of RAW264.7 cells were evaluated by MTT assay to exclude the interference of the cytotoxicity of the test compounds. All the tests were repeated three times.

### 2.8. Cytotoxicity and MDR Reversal Assay

The cisplatin (DDP) sensitive A549 and resistant A549/DDP cells purchased from IMMOCELL (Xiamen, Fujian, China) were cultured in Roswell Park Memorial Institute (RPMI-1640) or DMEM supplemented with 10% FBS. The tested compounds were prepared at 100 μg/mL DMSO stocks and diluted with fresh RPMI-1640 medium to final concentrations at 50 μg/mL and 100 μg/mL. Before determining MDR reversal activity, the viability of A549 and A549/DDP cells were evaluated by MTT assay to exclude the interference of the cytotoxicity of compounds **1**–**7**. A549 and A549/DDP cells were seeded in 96-well plates (5 × 10^4^ cells/well) for 24 h, and then the medium containing the tested compounds was added (100 μL/well) and cultured for another 24 h. The absorbance at 490 nm was measured. The MDR reversal activities were assayed via combining cisplatin (20 μg/mL) and the tested compounds. The verapamil was used as a positive control. All experiments were performed in parallel three times.

### 2.9. Statistical Analysis

All experiments were performed in triplicates and expressed as mean ± standard deviation (SD). An unpaired *t*-test was performed for data analyses using the GraphPad Prism software (version 5). *p* < 0.05 indicated a significant difference.

## 3. Results and Discussion

### 3.1. Structure Identification of Compounds ***1***–***7***

Biscogniauxiaol A (**1**) was isolated as a colorless solid with the molecular formula C_15_H_22_O_3_ determined by the HRFABMS (Appendix A), indicating five degrees of unsaturation. The infrared (IR) (Appendix A) spectrum of **1** displayed characteristic absorption bands for hydroxy (3438 cm^−1^) and double bond (1620 cm^−1^) functionalities. The ^1^H nuclear magnetic resonance (NMR) (Appendix A) data (Table 1) showed signals of three singlet methyls at δ_H_ 1.08 (H-15), 1.15 (H-12) and 1.16 (H-11), one D_2_O-exchangeable proton at δ_H_ 4.67 (10-OH), and two olefinic protons at δ_H_ 5.60 (1H, d, *J* = 5.7 Hz, H-2) and δ_H_ 5.93 (1H, d, *J* = 5.8 Hz, H-1). Followed by the interpretation of the ^13^C NMR (DEPT) (Appendix A) spectrum (Appendix A), fifteen carbon resonances included three methyls, four methylenes (one ether oxygen at δ_C_ 71.1), four methines (two olefinic carbons at δ_C_ 130.7, 147.0) and four oxy-nonprotonated carbons. The presence of one double bond accounted for one of the five degrees of unsaturation, suggesting that compound **1** required a tetracyclic skeleton (rings A–D).

The planar skeleton of **1** was constructed by interpretation of 2D NMR spectra (Figure 2). In the heteronuclear multiple bond correlation (HMBC) (Appendix A) spectrum, the correlations from the active hydrogen 10-OH to C-1/C-9/C-10/C-11 and from H-2 to C-1/C-3/C-9/C-10 suggested the presence of a five-membered cycloolefin structure (ring A) with a hydroxyl at C-10. The correlations from H_3_-12 to C-2/C-3/C-4/C-5 confirmed that C-4 was adjacent to C-3 and C-5. A spin system from C-5 to C-9 was established based on the ^1^H-^1^H COSY (Appendix A) correlations between H-5/H-6/H-7/H-8/H-9. Meanwhile, the connection between C-3 and C-9 was confirmed by HMBC correlations from H-5 to C-3/C-4/C-6/C-7 and from H-9 to C-3/C-10/C-8/C-4/C-11. The above data showed that C-3 to C-9 constructed a seven-membered ring structure. Further analyses of the HMBC correlations from H_3_-15 to C-7, C-13 and C-14 indicated that C-3 was adjacent to C-7. The HBMC correlations from H_3_-14 to C-7, C-13 and C-15 confirmed that C-3 was connected to C-14. Based on the HMBC correlations from H-14 to C-3, chemical shifts of C-4 (δ_C_ 76.5), C-3 (δ_C_ 85.7), C-13 (δ_C_ 72.1) and C-14 (δ_C_ 71.1) and the molecular formula of **1**, two ether bridges between C-4 and C-13, and C-3 and C-14 were determined. Thus, the planar structure and [5,6,6,7] ring skeleton of **1** were confirmed (Figure 1).

The relative configuration of **1** was established by analysis of the rotating-frame nuclear Overhauser effect spectroscopy (ROESY) (Appendix A) spectrum (Figure 3). The ROESY interactions of H-7/H-9/H_3_-12/H_3_-15 and H-9/10-OH indicated that these protons had the same orientation. The absolute configuration of **1** was determined by the time-dependent density-functional theory (TD-DFT) quantum calculation of two possible isomers. As shown in Figure 4, the calculated electronic circular dichroism (ECD) spectrum of 3*R*,4*S*,7*R*,9*S*,10*R*,13*R*–**1** matched well with the experimental one, which assigned its absolute configuration. Finally, compound **1** was named Biscogniauxiaol A, which was the first reported example of a [5/7] bicyclic sesquiterpenoid with an unprecedented [5/6/6/7] ring system.

Biscogniauxiaol B (**2**) was obtained as a colorless solid with the molecular formula C_15_H_28_O_3_ determined by the HRESIMS (Appendix A), and a corresponding two degrees of unsaturation. The ^1^H NMR data (Table 1) showed signals of three methyls at δ_H_ 0.82 (3H, d, *J* = 7.2 Hz, H-14), δ_H_ 0.88 (3H, d, *J* = 7.0 Hz, H-15), and δ_H_ 1.29 (3H, s, H-11). The ^13^C NMR (DEPT) (Appendix A) spectrum (Table 1) displayed 15 carbon resonances, which were recognized as three methyls, five methenes and six methines. The planar skeleton of **2** was constructed by interpretation of 2D NMR spectra (Figure 2). The COSY (Appendix A) correlations between H-1/H-2/H-3/H-4/H-5 and from H_3_-15 to H-4 constructed a five-membered ring structure (A). Furthermore, the COSY correlations between H-5/H-6/H-7/H-8/H-9 and the HMBC (Appendix A) correlations from H-9 to C-10, C-14 and C-1 constructed a seven-membered ring structure (B). The COSY correlations between H-7/H-11/H-12/H-13 suggested the presence of a isobutanol moiety at C-7. Finally, the planar structure of **2** was determined as shown in Figure 1. Its relative configuration was established by careful analyses of the ROESY (Appendix A) spectrum (Figure 3). The ROESY correlations H-1/H-4, H-1/H-7, H-7/H-5, H-1/H_3_-14 and H_3_-14/H-2 indicated that these protons were cofacial. Its absolute configuration was determined by the time-dependent density-functional theory quantum calculation of two possible isomers. The calculated electronic circular dichroism (ECD) spectrum of 1*S*,2*R*,4*S*,5*S*,7*R*,10*S*–**2** fitted well with the experimental one (Figure 4), which determined its absolute configuration.

Biscogniauxiaol C (**3**) was purified as a colorless solid with the molecular formula C_15_H_28_O_4_ determined by the HRESIMS (Appendix A), suggesting two degrees of unsaturation. A detailed comparison of its ^1^H (Appendix A) and ^13^C NMR (Appendix A) spectroscopic data (Table 1) with that reported for bicyclic sesquiterpene [25] indicated that **3** was tetrahydroxy derivatives of guaiane-type sesquiterpene. The ^13^C NMR (DEPT) (Appendix A) spectra of **3** displayed fifteen carbons consisting of three methyls, five methylenes of which one was oxygenated at δ_C_ 68.61, five methines of which one was oxygenated at δ_C_ 74.26 and two quaternary carbons that were oxygenated at δ_C_ 75.37 and 76.59. The analyses of COSY (Appendix A) and HMBC spectra (Appendix A) confirmed that **3** contained a guaiane-type sesquiterpene skeleton. The connection between C-11 and C-7 was confirmed by the HMBC correlations from H-7 to C-11. These data indicated that **3** shared the same planar structure as that of 1*R*,3*S*,4*R*,5*S*,7*R*,10*R*,11*S*-guaiane-3,10,11,12-tetraol [25]. However, comparison of their optical rotations and NMR data suggested that they were not identical but stereoisomeric. The relative configuration of 3 was assigned by the ROESY (Appendix A) spectrum (Figure 3). The ROESY correlations of H_3_-15/H_3_-14, H_3_-14/H-7 and H-7/H_3_-13 established that they were cofacial. Meanwhile, correlations between H-4 to H-1/H-3/H-5 indicated that these protons were on the same orientation of the molecule. The absolute configuration of 3 was determined by SR calculation of two possible isomers [26]. The calculated SR value for 1*S*,3*S*,4*S*,5*R*,7*S*,10*S*,11*S*–3 at the B3LYP/TZVP level was nearly consistent with the experimental SR value
$[\alpha {]}_{D}^{20}$−10.78 (*c* 0.10, MeOH) (Appendix A), which assigned the absolute configuration of **3** as 1*S*,3*S*,4*S*,5*R*,7*S*,10*S*, and 11*S*.

Biscogniauxiaol D (**4**) was obtained as a colorless solid with the molecular formula C_15_H_28_O_3_ established by the HRESIMS (Appendix A), implying two degrees of unsaturation. The ^13^C NMR (DEPT) (Appendix A) spectra of **4** (Table 2) revealed the presence of fifteen carbons consisting of three methyls, five methylenes of which one was oxygenated at δ_C_ 68.55, five methines of which one was oxygenated at δ_C_ 75.33 and two quaternary carbons that were oxygenated at δ_C_ 75.60 and δ_C_ 76.47. Its ^1^H and ^13^C NMR spectroscopic data were similar to that of compound **3**, indicating that compound **4** was a tetrahydroxy derivative of guaiane-type sesquiterpenes. The COSY (Appendix A) correlations of H-1 to oxidic H-2 and H-4 to H-3 suggested that the C-2 rather than C-3 was oxidized. In the analyses in combination with HMBC (Appendix A) and COSY spectrum correlations, the planar structure of compound **4** was elucidated as shown (Figure 1). The relative configuration of **4** was assigned by the ROESY (Appendix A) spectrum (Figure 3). The ROESY correlations H-1/H_3_-14, H_3_-14/H-2, H-2/H-5/H_3_-15, H-5/H-7 and H-7/H_3_-13 established that they were cofacial. Its absolute configuration was also determined by SR calculation of two possible isomers. The calculated SR value of 1*R*,2*R*,4*R*,5*R*,7*S*,10*S*–**4** at B3LYP/def2-TZVP level matched well with the experimental SR value
$[\alpha {]}_{D}^{20}$ −1.88 (*c* 0.12, MeOH) (Appendix A). Thus, its absolute configuration was determined as 1*R*,2*R*,4*R*,5*R*,7*S*, and 10*S*.

Biscogniauxiaol E (**5**) was isolated as a colorless solid and had the molecular formula C_15_H_28_O_4_ established by the HRESIMS (Appendix A), indicating 2 degrees of unsaturation. Its ^1^H and ^13^C NMR spectroscopic data were similar to that of compound **3**, suggesting that compound **5** was a tetrahydroxy derivative of guaiane-type sesquiterpenes. The ^13^C NMR (DEPT) (Appendix A) spectra of **5** (Table 2) had fifteen carbons consisting of three methyls, five methylenes of which one was oxygenated at δ_C_ 68.77, five methines of which one was oxygenated at δ_C_ 81.10 and two oxygenated carbons at δ_C_ 75.90 and 78.17. The COSY (Appendix A) correlations of H-9 to H-8 and the HMBC (Appendix A) correlations of H-14 to C-10 and H-13 to C-11 and C-12 confirmed that four hydroxyls were located at C-9, C-10, C-11 and C-12. The relative configuration of **5** was assigned by the ROESY (Appendix A) spectrum (Figure 3). The ROESY correlations H-1/H-9 and H-9/H-7 established that they were cofacial. Meanwhile, the ROESY correlations H_3_-14/H_3_-15/H-5 indicated that they were cofacial. The absolute configuration of **5** was determined by the time-dependent density-functional theory quantum calculation of two possible isomers. As shown in Figure 4, the calculated electronic circular dichroism (ECD) spectrum of the 1*S*,4*R*,5S,7*R*,9*R*,10*R*–**5** matched well with the experimental one, which assigned its absolute configuration.

Biscogniauxiaol 6 (**F**) was isolated as a colorless solid and its molecular formula C_15_H_28_O_4_ was established by the HRESIMS (Appendix A), corresponding to two degrees of unsaturation. The ^13^C NMR (DEPT) (Appendix A) spectra of **6** (Table 2) had fifteen carbons consisting of three methyls, five methylenes of which one was oxygenated at δ_C_ 68.54, five methines of which one was oxygenated at δ_C_ 73.90 and two quaternary carbons oxygenated at δ_C_ 74.62 and 75.82. Careful analyses of the ^1^ H NMR (Appendix A) and HSQC (Appendix A) data revealed that **6** had three methyl groups (δ_H_ 1.01, δ_C_ 10.01; δ_H_ 1.04, δ_C_ 18.43; δ_H_ 1.19, δ_C_ 30.92), five methylene groups of which one was oxygenate (δ_H_ 3.38 and δ_H_ 3.56, δ_C_ 68.54), and five methine groups of which one was oxygenate (δ_H_ 4.15, δ_C_ 73.90). The COSY (Appendix A) correlations of H-2 to H-1/H-3 and the HMBC (Appendix A) correlations of H-14 to C-10 and H-13 to C-11 and C-12 confirmed that four hydroxyls were located at C-2, C-9, C-11 and C-12 (Figure 2). The relative configuration of **6** was assigned by the ROESY (Appendix A) spectrum (Figure 3). The ROESY correlations H-7/H-5, H-5/H-1 and H-1/H_3_-15 established that they were cofacial. The ROESY correlations H-4/H-2 and H-4/H_3_-14 established that they were cofacial. The absolute configuration of **6** was also determined by SR calculation of two possible isomers. The SR value of the 1*S*,2*R*,4*S*,5*S*,7*S*,9*R*,11*R*–**6** agreed with the experimental SR value
$[\alpha {]}_{D}^{20}$−11.30 (*c* 0.10, MeOH) (Appendix A), which assigned its absolute configuration.

Biscogniauxiaol G (**7**) was obtained as a colorless solid with the molecular formula C_15_H_26_O_3_ deduced by the HRESIMS (Appendix A), indicating three degrees of unsaturation. Its ^1^H (Appendix A) and ^13^C NMR (Appendix A) spectroscopic data were similar to those of compounds **2** and **3**. The ^13^C NMR (DEPT) (Appendix A) spectra of **7** (Table 2) had fifteen carbons consisting of two methyls, six methylenes of which one was oxygenated at δ_C_ 65.09, five methines of which one was oxygenated at δ_C_ 78.75. The HMBC (Appendix A) correlations H_3_-14 to C-10 and H-12 to C-11/C-13 and their chemical shift confirmed that three hydroxyls were located at C-3, C-10 and C-12 (Figure 2). The ^13^C NMR data and HMBC correlations demonstrated the presence of two sp^2^ carbons (δ_C_ 107.51 and 157.27) at C-11 and C-13. The relative configuration of **7** was assigned by the ROESY (Appendix A) spectrum (Figure 3). The ROESY correlations of H-1/H-3, H-3/H_3_-15, H-7/H-5 and H-5/H_3_-15 indicated that they were cofacial. The absolute configuration of **7** was determined by the time-dependent density-functional theory quantum calculation of two possible isomers. As shown in Figure 4, the ECD spectrum of the 1*S*,3*R*,4*R*,5*R*,7*S*,10*S*–**7** matched with the experimental one, which allowed the assignment of the absolute configuration of **7** as 1*S*,3*R*,4*R*,5*R*,7*S*, and 10*S*.

It was acknowledged that farnesyl diphosphate (FPP), the precursor of guaiane-type sesquiterpenoids, was substituted by various hydroxyls via oxidation reactions to afford oxidized guaiane-type sesquiterpenoids [27]. Therefore, a plausible biosynthetic pathway for **1**–**7** was proposed (Figure 5).

### 3.2. Results of Bioactivity Assays

#### 3.2.1. Anti-Fungal Evaluation of Compounds

In consideration of the previously discovered anti-fungal activities for guaiane-type sesquiterpenoids [28], the inhibitory effects of compounds **1**–**7** against *C. albicans* (336485) were evaluated. As shown in Table 3, all compounds had inhibitory activities against *C. albicans*. Among them, compounds **1**, **2**, and **7** exhibited as potent with MICs of 1.60, 6.25 and 6.30 μM, respectively (Amphotericin B and Fluconazole with MICs of 0.43 and 2.61 μM, respectively).

#### 3.2.2. Anti-Inflammatory Activities of Compounds

The anti-inflammatory is one of the most important biological activities for guaiane-type sesquiterpenoids [29]. The inhibitory effects of compounds **1**–**7** on the LPS-induced production of NO in the RAW264.7 cell line were evaluated in vitro. The results showed that compounds **1**, **2**, and **7** suppressed the NO production with IC_50_ values of 4.60 ± 0.42, 20.00 ± 1.54 and 18.38 ± 1.12 μM, respectively (indomethacin, IC_50_ = 22.94 ± 1.42 μM), and none of the compounds exhibited cytotoxicity (Table 4).

#### 3.2.3. Anti-cancer and MDR Reversal Effects of Compounds

According to literature reports, guaiane-type sesquiterpenoids had anticancer and MDR reversal activities [30,31]. Thus, the anti-cancer and reversal activities of compounds **1**–**7** in the cisplatin sensitive A549 cells and resistant A549/DDP cells were evaluated in vitro, respectively. The data displayed that all compounds showed no cytotoxicity to the cisplatin sensitive A549 cells and resistant A549/DDP cells at concentration of 100 μg/mL (Figure 6). However, they had weak reversal activities to the cisplatin resistant A549/DDP cells at concentrations of 50 μg/mL and 100 μg/mL compared to the control group DDP (Figure 7 and Figure 8).

## 4. Conclusions

In summary, seven new bioactive sesquiterpenoids including an unprecedented [5/6/6/7] tetracyclic system iridoid were obtained from the endophytic fungus viz *B. petrensis* on *D. orchids*. Their structures including absolute stereochemistry were determined. A possible biosynthetic pathway for them was proposed, which may promote further chemical synthesis efforts. Compounds **1**, **2**, **7** showed potent inhibitory effects on *C. albicans* with MIC values of 1.60, 6.25 and 6.30 μM, respectively. Meanwhile, they exhibited prominent inhibitory activities against the NO production in the RAW264.7 cells with IC_50_ values of 4.60 ± 0.42, 20.00 ± 1.54, 18.38 ± 1.12 μM, respectively. Moreover, all the isolated compounds had weak reversal activities to the cisplatin resistant A549/DDP cells. These data confirmed that the endophytic fungus *B. petrensis* is a new source of bioactive guaiane-type sesquiterpenoids that can replace the plants, and compounds **1**, **2** and **7** (biscogniauxiaols A, B and G) were promising for further optimization as multifunctional inhibitors for anti-fungal (*C*. *albicans*) and anti-inflammatory purposes.

## Figures and Tables

**Figure 1 jof-09-00393-f001:**
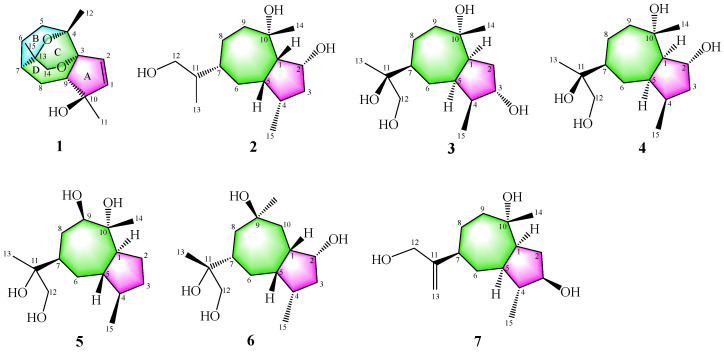
The structures of compounds **1**–**7**.

**Figure 2 jof-09-00393-f002:**
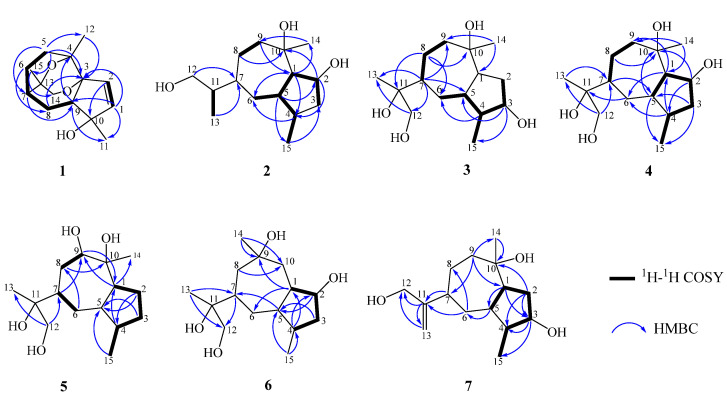
Key ^1^H–^1^H COSY and HMBC of compounds **1**–**7**.

**Figure 3 jof-09-00393-f003:**
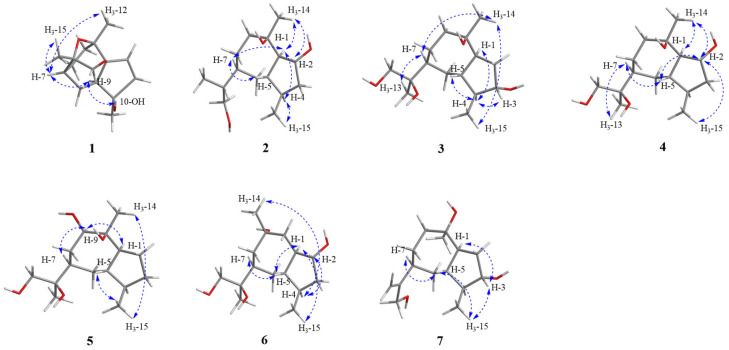
Key ROESY correlations of compounds **1**–**7**.

**Figure 4 jof-09-00393-f004:**
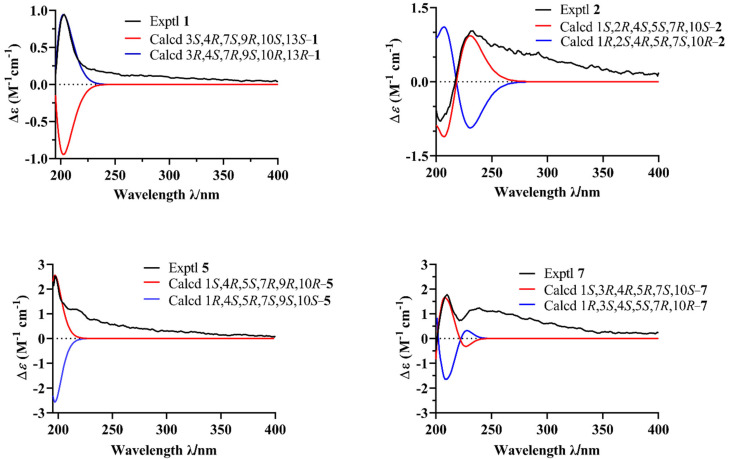
Experimental and calculated ECD spectrum of compounds **1**, **2**, **5**, and **7**.

**Figure 5 jof-09-00393-f005:**
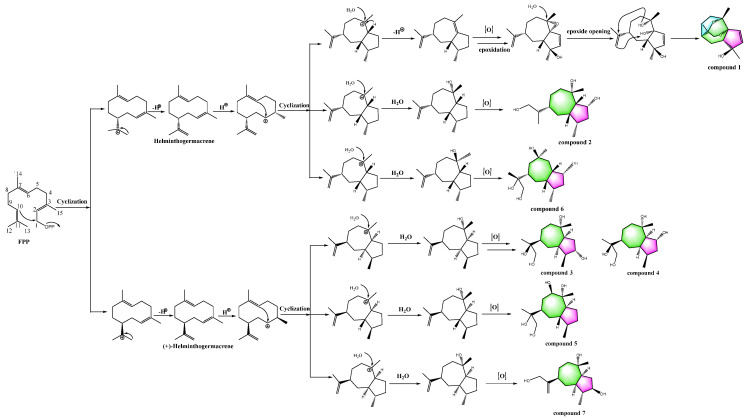
Plausible biosynthetic pathway of compounds **1**–**7**.

**Figure 6 jof-09-00393-f006:**
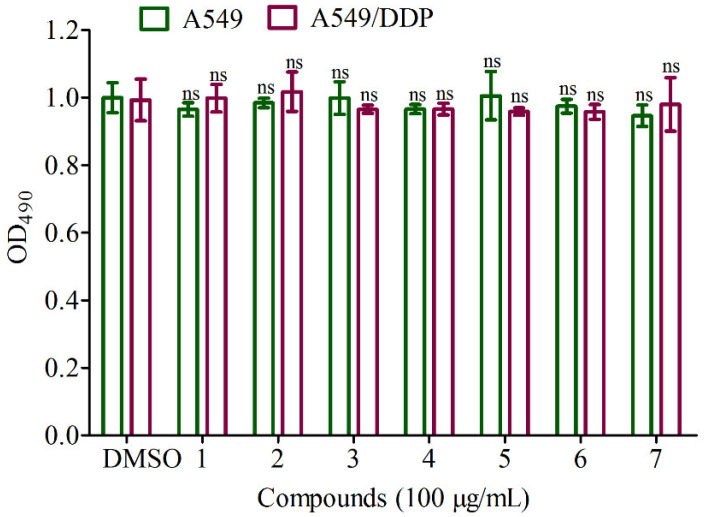
Inhibition effects of the isolates at concentration of 100 μg/mL on A549 and A549/DDP. ^ns^ *p* > 0.05 compared to the negative group (DMSO).

**Figure 7 jof-09-00393-f007:**
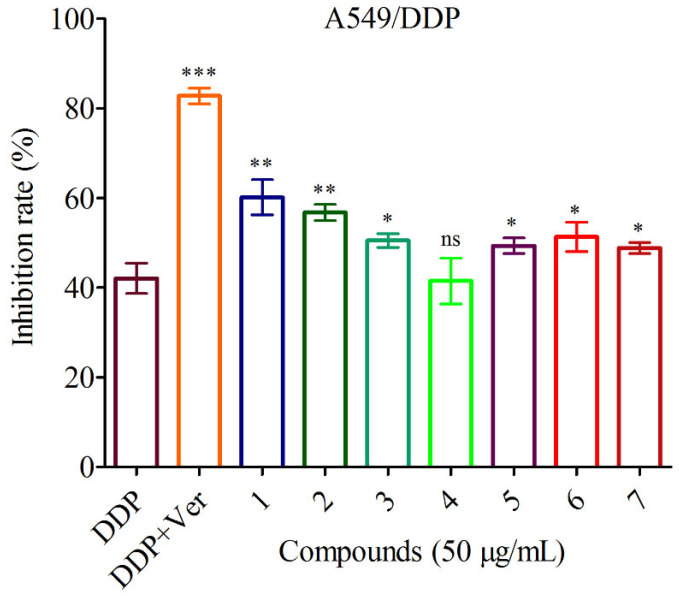
Reversal activities of the isolates at concentration of 50 μg/mL against A549/DDP. DDP: cisplatin; Ver: the positive drug verapamil. *** *p* < 0.001, ** *p* < 0.01, * *p* < 0.05 and ^ns^
*p* > 0.05 compared to the control group (DDP), respectively. The following are the same.

**Figure 8 jof-09-00393-f008:**
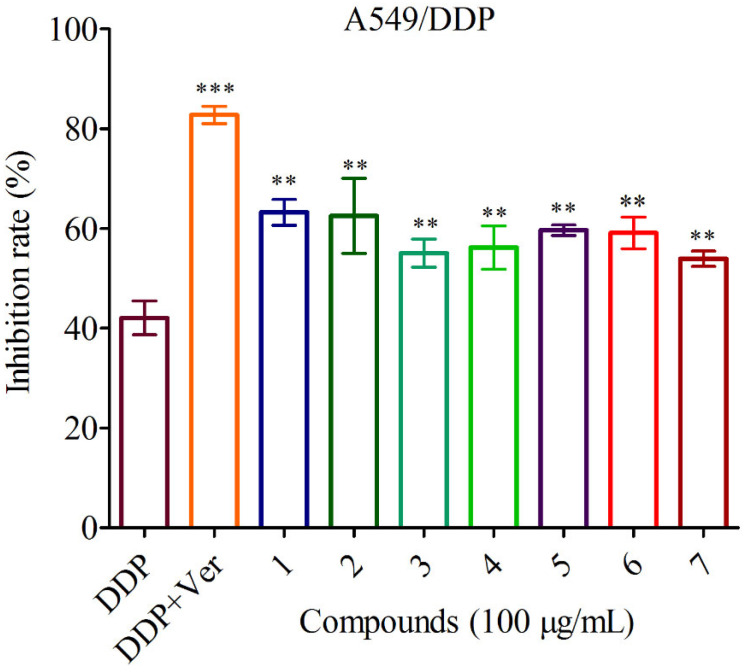
Reversal activities of the isolates at concentration of 100 μg/mL against A549/DDP.

**Table 1 jof-09-00393-t001:** ^1^H (500 MHz) and ^13^C (125 MHz) NMR data for compounds **1**−**3** (*δ* in ppm, *J* in Hz).

No.	1 (In DMSO-*d6*)		2 (In CD_3_OD)		3 (In CD_3_OD)	
	δ_C_	δ_H_	δ_C_	δ_H_	δ_C_	δ_H_
1	147.0, CH	5.93 d (5.7)	62.75, CH	2.03 dd (9.8, 7.5)	52.72, CH	2.06 m
2	130.7, CH	5.60 d (5.8)	75.33, CH	4.23 q (7.0)	37.04, CH_2_	2.09 m 1.53 m
3	85.7, C		43.10, CH_2_	1.66 m	74.26, CH	4.07 m
4	76.5, C		37.15, CH	2.18 m	44.75, CH	1.94 m
5	32.3, CH_2_	1.61 m 1.43 m	46.50, CH	2.27 m	45.88, CH	2.02 m
6	22.5, CH_2_	1.97 m 1.54 m	28.50, CH_2_	1.31 m 1.05 m	27.58, CH_2_	1.57 m 1.19 m
7	36.6, CH	1.87 m	41.14, CH	1.66 m	46.75, CH	1.69 m
8	26.9, CH_2_	1.83 m 1.5 m	30.01, CH_2_	1.63 m 1.38 m	25.44, CH_2_	1.91 m 1.34 m
9	56.2, CH	2.44 dd (12.2, 8.0)	41.85, CH_2_	1.82 m	39.24, CH_2_	1.92 m 1.52 m
10	80.1, C		75.72, C		75.37, C	
11	23.3, CH_3_	1.16 s	43.47, CH	1.54 m	76.59, C	
12	25.7, CH_3_	1.15 s	66.29, CH_2_	3.46 dd (10.8, 6.7) 3.33 dd (10.8, 7.1)	68.61, CH_2_	3.44 d (7.1)
13	72.1, C		12.88, CH_3_	0.82 d (7.0)	20.81, CH_3_	1.07 s
14	71.1, CH_2_	3.87 d (8.9) 3.27 d (8.9)	28.25, CH_3_	1.29 s	28.58, CH_3_	1.18 s
15	25.0, CH_3_	1.08 s	16.79, CH_3_	0.88 d (7.2)	10.15, CH_3_	0.92 d (7.3)
10-OH	-OH	4.67 s				

**Table 2 jof-09-00393-t002:** ^1^H (500 MHz) and ^13^C (125 MHz) NMR data for compounds **4**−**7** (*δ* in ppm, *J* in Hz).

No.	4 (In CD_3_OD)		5 (In CD_3_OD)		6 (In CDCl_3_)		7 (In CD_3_OD)	
	δ_C_	δ_H_	δ_C_	δ_H_	δ_C_	δ_H_	δ_C_	δ_H_
1	63.06, CH	2.01 m	50.34, CH	2.16 m	53.38, CH	2.00 m	49.84, CH	2.23 m
2	75.33, CH	4.19 m	27.52, CH_2_	1.81 m	73.90, CH	4.15 m	36.99, CH_2_	2.09 m 1.58 m
3	43.19, CH_2_	1.64 m 1.70 m	33.78, CH_2_	1.60 m 1.39 m	37.14, CH_2_	2.23 m 1.48 m	78.75, CH	3.44 m
4	37.28, CH	2.19 m	40.09, CH	2.06 m	43.77, CH	1.97 m	49.52, CH	1.24 s
5	47.14, CH	2.25 m	46.13, CH	2.12 m	45.87, CH	1.95 m	47.20, CH	1.60 m
6	27.45, CH_2_	0.99 m 1.55 m	29.82, CH_2_	1.63 m 1.09 m	24.64, CH_2_	1.77 m 1.19 m	38.52, CH_2_	1.75 m 1.44 m
7	46.40, CH	1.72 m	44.58, CH	1.59 m	44.92, CH	1.75 m	44.54, CH	1.97 m
8	25.53, CH_2_	1.32 m 1.91 m	34.90, CH_2_	2.20 m 1.30 m	23.69, CH_2_	1.77 m 1.17 m	33.34, CH_2_	1.73 m 1.51 m
9	40.49, CH_2_	1.61 m 1.85 m	81.10, CH	3.37 m	74.62, C		46.51, CH_2_	1.86 m 1.67 m
10	75.60, C		78.17, C		33.86, CH_2_	1.95 m 1.52 m	75.95, C	
11	76.47, C		75.90, C		75.82, C		157.27, C	
12	68.55, CH_2_	3.43 d (3.7)	68.77, CH_2_	3.44 m	68.54, CH_2_	3.38 d (10.95) 3.56 d (10.95)	65.09, CH_2_	4.01 s
13	21.08, CH_3_	1.06 s	20.60, CH_3_	1.06 s	18.43, CH_3_	1.04 s	107.51, CH_2_	4.95 m 4.82 s
14	29.08, CH_3_	1.29 s	18.43, CH_3_	1.14 s	30.92, CH_3_	1.19 s	23.70, CH_3_	1.25 s
15	16.66, CH_3_	0.90 d (7.0)	16.04, CH_3_	0.83 d (7.1)	10.01, CH_3_	1.01 d (6.95)	16.34, CH_3_	0.96 d (6.4)

**Table 3 jof-09-00393-t003:** The inhibitory activities of compounds **1**–**7** against *C.albicans*.

Compounds	MIC (μM)
**1**	1.60
**2**	6.25
**3**	23.53
**4**	12.50
**5**	11.76
**6**	47.06
**7**	6.30
Amphotericin B *^a^*	0.43
Fluconazole *^b^*	2.61

*^a^* Amphotericin B and *^b^* Fluconazole: the positive controls.

**Table 4 jof-09-00393-t004:** The inhibitory effects of compounds **1**–**7** on NO production in LPS-stimulated RAW264.7.

Compounds	IC_50_ (μM)	CC_50_ (μM)
**1**	4.60 ± 0.42	>80
**2**	20.00 ± 1.54	>100
**3**	60.20 ± 0.81	>80
**4**	62.48 ± 1.23	>100
**5**	50.76 ± 0.68	>100
**6**	75.50 ± 0.73	>100
**7**	18.38 ± 1.12	>100
Indomethacin ^a^	22.94 ± 1.42	>100

^a^ Indomethacin: the positive control.

## Data Availability

Data supporting the reported results are provided in Appendix A. The data from manuscript and Appendix A are available for publication, citation, and use.

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
