# Peer review of "New Guaiane-Type Sesquiterpenoids Biscogniauxiaols A–G with Anti-Fungal and Anti-Inflammatory Activities from the Endophytic Fungus Biscogniauxia Petrensis"

_jof, 2023, doi:10.3390/jof9040393_

Round 1

Reviewer 1 Report

The manuscript is quite interesting, as well as, the isolated substances, mainly compound 1. The isolated molecules showed relevant biological activities. Appropriate techniques were used for structural elucidation, such as NMR 1 and 2D and high-resolution mass spectrometry. In the text there are some corrections that should be done. 

In line 79, a comment was inserted about  fungus cultivation and obtaining the extracts that could have been better explained.

The attached file shows some yellow marks to help with corrections.

Beware of the units as they have been modified.

Check the guideline for authors to standardize the references. The abbreviations of the journals are not standardized, some have a period, others do not. One reference has no title.

The manuscript is well written and the results are relevant.

Author Response

Point 1: in line 79, a comment was inserted about fungus cultivation and obtaining the extracts that could have been better explained.

Response 1: Thanks for referee’s kind suggestions. we have inserted the contents about fungus cultivation and obtaining the extracts in line 77-81 the revised manuscript. Martin modified (MM) medium was inoculated with the aforementioned fungus and incubated in a constant-temperature incubator at 28 ℃ for 5 d to obtain seed culture. Fermentation was carried out in a conical flask (1 L) containing 200 g of rice and 150 mL of distilled water, and then autoclaving at 120 ℃ for 30 minutes. Each flask was inoculated with 10 mL of seed culture and incubated at 28 â„ƒ for two months. The fermented cultures were extracted thrice with 2-fold volume of methanol and the organic solvent was evaporated to a small volume under vacuum, and then suspended in H2O (10 L) and partitioned successively with 20 L of ethyl acetate and n-Butanol respectively. The ethyl acetate solution was evaporated under reduced pressure to afford a crude extract (56 g).

Point 2: The attached file shows some yellow marks to help with corrections.

Response 2: Thanks for referee’s advice. we have amended the contents in yellow marks, such as μM, μg/mL and Manufacturer's citation abbreviation et al in the revised manuscript.

Point 3: Be ware of the units as they have been modified.

Response 3: Thanks for referee’s kind reminder. We are aware of this problem. We will pay attention to it next time when we upload the file.

Point 4: Check the guideline for authors to standardize the references. The abbreviations of the journals are not standardized, some have a period, others do not. One reference has no title.

Response 4: Thanks for referee’s suggestions. we have amended the nonstandard abbreviations of the journals in the revised manuscript. Moreover, we have revised the reference 6 in the manuscript “Wu, S.H.; He, J.; Li, X.N.; Huang, R.; Song, F.; Chen, Y.W.; Miao, C.P. Guaiane sesquiterpenes and isopimarane diterpenes from an endophytic fungus Xylaria sp. Phytochemistry. 2014, 105, 197-204.” (Line 419-486)

Reviewer 2 Report

The authors reported seven undescribed guaiane-type sequiterpenoids from B. petrensis, structures were elucidated through a combination of NMR, spectroscopic techniques, et al. Three out of the seven compounds are found to be promising candidates for combating fungal infection and inflammatory. This work can be published in its current form.   

Author Response

Point 1: The authors reported seven undescribed guaiane-type sequiterpenoids from B. petrensis, structures were elucidated through a combination of NMR, spectroscopic techniques, et al. Three out of the seven compounds are found to be promising candidates for combating fungal infection and inflammatory. This work can be published in its current form.

Response 1: Thank you very much for your detailed review and positive comments on the manuscript.
